# Influence of the Carbidized Tungsten Surface on the Processes of Interaction with Helium Plasma

**DOI:** 10.3390/ma15217821

**Published:** 2022-11-05

**Authors:** Mazhyn Skakov, Arman Miniyazov, Erlan Batyrbekov, Viktor Baklanov, Yerbolat Koyanbayev, Aleksandr Gradoboev, Yernat Kozhakhmetov, Igor Sokolov, Timur Tulenbergenov, Gainiya Zhanbolatova

**Affiliations:** 1National Nuclear Center of the Republic of Kazakhstan, Kurchatov 071100, Kazakhstan; 2Institute of Atomic Energy Branch of the National Nuclear Center of the Republic of Kazakhstan, Kurchatov 071100, Kazakhstan; 3Faculty of Engineering and Technology, Shakarim University, Semey 070000, Kazakhstan; 4Department of Experimental Physics, Tomsk Polytechnic University, 634050 Tomsk, Russia

**Keywords:** ITER, divertor, tungsten, carbide layer, tokamak, helium plasma, microstructure, TEM

## Abstract

This paper presents the results of experimental studies of the interaction of helium plasma with a near-surface tungsten carbide layer. The experiments were implemented at the plasma-beam installation. The helium plasma loading conditions were close to those expected in the ITER divertor. The technology of the plasma irradiation was applied in a stationary type linear accelerator. The impact of the helium plasma was realized in the course of the experiment with the temperatures of ~905 °C and ~1750 °C, which were calculated by simulating heat loading on a tungsten monoblock of the ITER divertor under the plasma irradiation at the load of 10 MW/m^2^ and 20 MW/m^2^, respectively. The structure was investigated with scanning microscopy, transmitting electron microscopy and X-ray analysis. The data were obtained showing that the surface morphology changed due to the erosion. It was found that the carbidization extremely impacted the plasma–tungsten interaction, as the plasma–tungsten interaction with the carbide layer led to the carbon sputtering and partial diffusion towards to the depth of the sample. According to these results, WC-based tungsten carbide is less protected against fracture by helium than W and W_2_C. An increase in temperature leads to much more extensive surface damage accompanied by the formation of molten and recrystallized flanges.

## 1. Introduction

It is well known that tungsten chosen as a plasma-faced divertor material has a high melting point, good thermal conductivity and very high threshold for sputtering under the action of plasma particles and small tritium capture [1,2,3]. In fusion reactors, surfaces of plasma facing components (PFCs) are exposed to high heat and particle flux, both in stationary tokamak operation modes and in non-stationary peripheral plasma instabilities and plasma disruptions [4,5,6,7].

The effect of plasma and thermonuclear fusion products on tungsten leads to the formation of a crystal lattice and their complexes defects in the entire amount of the material, where the accumulation of helium atoms, hydrogen and its isotopes is possible. Moreover, the changes in the microstructure of the surface, recrystallization, erosion, sputtering, melting and cracking, redeposition of tungsten, formation of inhomogeneous and porous layers on the surface of tungsten are observed during the plasma-thermal load [8,9,10,11,12].

According to the expected conditions in the divertor region in ITER, tungsten will be exposed to helium ions with the energy from 10^1^ up to 10^4^ eV and an ion flux ≥ 10^22^ ion/m^2^s [8,9].

According to the research results of the last decade, it is known that on the tungsten surface, when exposed to helium plasma, a “fuzz” is formed—a layer of microscopic elements with a high content of voids on the surface, as well as helium blisters [13,14,15,16]. It is also known that the blisters in tungsten accumulate at the grain boundaries and dislocations, change the surface hardness and increase the hydrogen isotopes retention.

Based on the results of experiments in the simulated linear plasma benches, it has been found that the energy of incident ions, ion flux and surface temperature are the key parameters of the change in the morphology of the tungsten surface [15,16].

When tungsten is irradiated with low-energy helium ions (10 eV–1 keV), an increase in “fuzz” is observed. However, the ion flux should be = 10^22^ ion/m^2^s, and the surface temperature ~600 ÷ 1750 °C [13,15,16,17].

In paper [18], studies on the effect of carbon impurities on the morphology of the tungsten surface when irradiated with helium ions were carried out. The experiments on tungsten irradiation with the mixed helium and carbon ions (0.5%) with a total fluence of ~10^25^ m^−2^ at a temperature of 905 °C were carried out. According to the results of X-ray photoelectron spectroscopy, it was found that the sample surface was covered with a layer of carbon in a graphite form. The obtained carbon layer prevented the formation of “fuzz” on the sample surface.

However, such nanostructures are very fragile and easily detach from the surface under high thermal loads, representing a source of tungsten dust, which is a serious threat to lowering the plasma temperature in Tokamaks [19,20,21].

As the energy of helium ions increases (1 keV–300 keV), the changes in the morphology of the tungsten surface can be characterized by helium blisters, flakes and pores. The formation depth and density as well as the blisters size depend on the values of the ion flux and the temperature of the sample [22,23,24].

An increase in the surface temperature during the irradiation with an ion flux of 10^18^–10^21^ ion/m^2^s leads to the fact that most of the blister caps peel off, leaving a porous surface.

Most fusion plants use either tungsten coatings applied to graphite and carbon–graphite materials or uncoated graphite materials, such as in the Kazakhstani material-testing tokamak KTM [25]. The presence of various materials in the chamber of the thermonuclear installations, as well as their impurities, as a result of erosion, will lead to the formation of mixed layers in plasma-facing surfaces during the re-deposition, in particular the carbide layer [26]. At the same time, the effect of the formation of carbide layers on the change in surface morphology during the interaction of plasma with tungsten, in our opinion, has not been fully studied.

The carbide layer on the tungsten surface can act as a diffusion barrier and plays an important role in the capture, retention and diffusion of helium and hydrogen ions into tungsten [26,27,28,29,30]. In [29], to study the diffusion of helium and carbon into tungsten, tungsten was irradiated with a mixed beam of helium and carbon ions. It is assumed that the internal gas pressure due to trapped helium and its accumulation at the boundary of vacancies or grains of tungsten carbides can deform the surface layer and form blisters. It was found that the amount of carbon impurity has a great influence on the blisters’ formation. It is also possible that the implanted carbon diffuses beyond the range of helium ions and creates defective areas where helium is trapped. However, an increased carbon concentration leads to a carbon film formation. With such irradiation, the formation of blisters on the surface of tungsten was not observed.

It was found in [27,31,32,33,34] that carbon impurities, in the form of graphene, can significantly slow down the changes in the morphology of the tungsten surface bombarded with helium and hydrogen ions in a wide range of energies.

Thus, based on the analysis of the literature, it was established that the key parameters influencing changes in the morphology of the tungsten surface are the energy of the incident ions, ion flux and surface temperature. It was also found that when tungsten is exposed by the helium plasma, the surface modification occurs with the formation of “fuzz” and blisters.

Based on the above, the influence of the carbide layer on the interaction processes of the plasma with the plasma-turned surface of tungsten is obvious. Therefore, the purpose of this work is to experimentally investigate the influence of the carbide layer on the interaction processes of plasma with the tungsten surface.

## 2. Materials and Methods

In determining the pristine surface features of tungsten, samples that had not undergone surface carbidization (samples No.1 and No.2) were used for re-testing. The total number of tungsten samples in accordance with the results presented in this paper was 6 pcs, including 2 samples with a WC carbide layer and 2 samples with a W_2_C carbide layer. According to the results of our recent work [35], the thickness of the carbide layer was in the range from 1.6 to 4.2 µm. The tungsten samples were discs cut from a bar of the highest purity (HP) with a height of 2.0 ± 0.1 mm and a diameter of 10 mm (S = 0.78 cm^2^) via the electroerosion method. All the samples were previously subjected to recrystallization annealing. Figure 1 shows the appearance of the tungsten samples with a carbidized surface. The marking and order of the samples were indicated in accordance with the phase composition of the surface layer of the samples.

To achieve this goal, the experiments were first carried out to obtain a carbide layer on the surface of the tungsten samples. In accordance with the method previously developed by us [36], the carbide surface layers were obtained in the form of WC tungsten carbide and W_2_C tungsten semi-carbide. The results of the tungsten carbidization process studies are described in [35,37].

The phase composition of the surface layers in samples 3 and 4 consisted of WC, while in samples 5 and 6 it consisted of W_2_C. After the carbidization on all the samples, the surface, regardless of the experimental conditions, was characterized by the absence of metallic luster characteristics in the initial samples. The surfaces of the WC-3 and WC-4 samples were characterized by the presence of a dark shade layer, while they had visually detectable matte zones. No other visible defects were found on the samples’ surfaces, with the exception of sample 4, where some large chips were observed at the edges, which were formed as a result of fixing the sample into the collector assembly of the experimental installation.

When examining samples with a layer of W_2_C tungsten semi-carbide, an intermediate area with a matte light gray tint was observed on the surface of the sample W_2_C-5. In this case, the central part of the sample was characterized by a visually observable grain structure with a diameter of about Ø 4.5 mm passing into an intermediate section with a matte area, with a width of about 1 mm (inner ~Ø 4.5 mm, outer ~Ø 5.5 mm). There was a transition back to a structure similar to the central section with a slight enlargement of the grain structure closer to the periphery. The study of the effect of the carbide layer on the processes of the interaction of the gel plasma with tungsten was accompanied by the materials science studies of the surface of the tungsten samples.

The nature of the damage to the tested samples was determined via visual inspection using a macro image of the surface.

The studied samples’ surface images were obtained using a Canon EOS 1200D camera. The microstructure and elemental composition of the samples were studied in the topographic and compositional contrast mode using a TescanVega3 scanning electron microscope with an X-Act energy dispersive spectrometer. The studies of the fine structure of the cross-section of the tungsten samples were performed using an HRTEM JEM-2100F (Joel, Japan, Tokyo) transmission electron microscope (TEM) at an accelerating voltage of 200 kV with a Schottky thermal field gun (Tomsk Polytechnic University, Tomsk, Russia). The roughness parameters were determined using the Mitutoyo Surftest SJ-410 profilometer on the working surface along the diametral groove.

X-ray diffraction patterns of the samples were taken on an Empyrean diffractometer in the PIXcel1D (scanning linear detector) operating mode. The processing of diffractograms was carried out by “HighScore” program for processing and searching. The processing of diffractograms was carried out by a program for processing and searching. To identify the phase composition, the Crystallography Open Database (hereinafter–COD) and the PDF-2 ICDD Release 2004 database were used.

## 3. Experimental Part

The conditions for conducting experimental work on a plasma-beam installation were determined to study the effect of helium irradiation on the morphology of the tungsten surface.

Earlier, we carried out work on modeling the thermal load on plasma-converted tungsten under the conditions of ITER regular operation [38]. Based on the simulation results, the temperature values for plasma irradiation were calculated to simulate a thermal load corresponding to heat flows in ITER, which was 905 °C (corresponds to ~10 MW/m^2^ stationary) and 1750 °C (corresponds to ~20 MW/m^2^ pulsed). The pressure in the chamber ranged from 1.03·10^−3^ Torr to 1.2·10^−3^ Torr. The helium ion flux was ≥10^22^ ion/m^2^s.

The experiments on the implementation of the parameters of the tungsten test were carried out on the plasma-beam installation [39,40,41]. The tungsten samples were installed in the target assembly, which was located in the interaction chamber. Figure 2 shows plasma-beam installation and the temperature control of the sample from the backside of the sample.

The thermal load on the surface of the samples was carried out together with the helium plasma action. Helium was supplied to the interaction chamber pre-pumped to a high vacuum to form a plasma-beam discharge using an electron beam. The data were recorded in real time. Table 1 shows the main parameters of the experiments.

The impact power of the plasma-beam discharge varied depending on the calculated temperature of the tungsten surface, as described above [38]. The ion current on the sample was measured by an ammeter of the power supply, which provided a potential supply of minus 2000 V. The current density was determined by placing the diaphragm in front of the target node with an aperture with a diameter of 10 mm.

Temperature control on the irradiated tungsten surface was carried out by a non-contact method using a two-wave pyrometer of the ISR6 brand. On the back side, the temperature was measured by the contact method using a tungsten–rhenium thermocouple of the WRh-5/20 type (Tungsten/rhenium-alloy thermocouples Type C).

During the irradiation, the contact diagnostic of helium plasma near the irradiated surface of tungsten was carried out using a Langmuir probe.

Table 2 shows the volt-ampere characteristics processing and calculation of the local parameters, which were carried out by the standard methods [42] using QtiPlot software for the analysis and visualization of the scientific data.

In the field of plasma exposure to tungsten, the contactless monitoring of plasma-beam discharge was also provided using the HR 2000+ ES optical spectrometer and OceanView software. The spectrometric analysis in the interaction chamber of the installation was carried out online at an integration time of 1000 ms. Figure 3 shows the graph of the optical spectra under two temperature modes.

Only high-intensity helium spectral lines can be observed on the graph. The identification of helium spectral lines was carried out according to the NIST Atomic Spectra Database [43].

It has been established that the intensity of the radiation of helium plasma depends on the irradiation modes, namely, it increases with an increasing in the electron beam power. This is due to an increase in the number of helium ions in the volume, which is confirmed by a higher ion current (Table 1).

## 4. Results and Their Discussion

### 4.1. Macrostructure of the Tungsten Surface after Irradiation with Helium Plasma

Figure 4 shows the macro images of the surface of the initial and carbidized tungsten samples after irradiation with helium plasma. The surface structure of the two samples differed significantly depending on the irradiation temperature and the samples were divided into two groups depending on the temperature of exposure.

The samples irradiated at a temperature of ~905 °C were characterized by a light gray matte shade with the presence of local areas with a grain structure (green arrows). Concerning sample No. 1, without a carbide layer, the structure had a clear outline in the form of a ring with a width of 1.5 mm (inner Ø = 5.5 mm and outer Ø = 7 mm), whereas the grain structure of sample No. 2 was characterized by a local characteristic arrangement.

The surfaces of the samples irradiated at ~1750 °C were characterized by a continuous grain structure, while the presence of large grains was observed on the periphery of the surface of samples No. 2 and No. 6. Sample No. 4 has a uniform fine-grained structure.

### 4.2. Microstructure of the Surface of the Initial Samples after the Irradiation with Helium Plasma

For the comparative analysis, two initial tungsten samples were irradiated with helium plasma at different temperatures (No. 1 ~905 °C and No. 2 ~1750 °C) at the initial stage. The chemical composition of the surface of the samples after irradiation was measured by EDS analysis (Table 3). The EDS method of the irradiated surface of both samples showed a characteristic pure W spectrum, in addition to a weak oxygen peak. We believe that the presence of oxygen on the sample surface was due to the adsorption of water, which may have occurred in the vacuum chamber after the He^+^ ion irradiation process.

The surface of sample No. 1 (Figure 5) irradiated at a temperature of ~905 °C was characterized by the formation of large micron blisters, the size of which reached 10 microns. At the same time, a high concentration of submicron helium blisters was observed in the body of the micron helium blisters. Another important feature of the microstructure of the tungsten sample surface was the presence of a wavy structure, as Figure 5 shows. In general, the formation of such a mixed structure (helium blisters and oriented waviness) on the sample No. 1 surface most likely depended on the tungsten grain orientation. This was confirmed by the fact that the shapes and sizes of both the large micron helium blisters and areas with a wavy structure almost completely echoed the outlines of the W grains and had clear polyhedral boundaries. According to the authors of paper [44], the polyhedral shape of the helium blisters indicated that the pressure of the gaseous helium inside the blisters was lower than that determined by the surface energy. That is, the lower the gas pressure inside the blister, the less flaking occurred. In our case, the opening of the helium blisters was almost not observed.

The irradiation of tungsten samples with helium plasma at a temperature of ~1750 ° C led to the formation of various surface damages on the surface of tungsten sample No. 2. The main feature of the sample surface structure was the presence of a developed coral structure with a large concentration of holes on their surfaces with a size of no more than 1 µm. At the same time, the morphology and orientation of the coral structure varied from grain to grain (Figure 6b,c, marked in green). The grain boundaries of tungsten had clear outlines and were characterized throughout the tungsten sample by large dimensions (±150 µm). The formation of helium blisters, the size of which varied in a wide range of values and could reach up to 20 µm, was observed in the body and at the grain boundaries of tungsten (Figure 6d,e, marked in red). According to the obtained results, it was found that the local large helium blisters were mainly concentrated in the body of the tungsten grains, whereas the smaller helium blisters accumulated at the boundaries. It is important to note that for the surfaces of helium, blisters are characterized by a coral structure without a specific morphology regardless of size.

The above experimental results showed the dependence of the morphology change on the crystallographic orientation of the grains. According to the results of work [45], tungsten grains are divided into two types depending on morphology:−crystal grains with orientation {1 0 1}, {0 0 1}, {1 1 2} and {1 1 1} have a structure with a large number of extended oriented protrusions;−crystal grains with orientation {1 0 3}, {1 0 2}, {4 0 7} and {2 0 3} are characterized by the presence of a bumpy structure with a large number of vertically upward oriented protrusions relative to the surface.

### 4.3. Microstructure of the Tungsten Samples Surface with a Carbide Layer Based on the WC Phase after Irradiation with Helium Plasma

Helium plasma irradiation of the tungsten samples with a WC-based carbide layer was carried out under similar conditions as for the initial samples, that is, at an ion energy of 2 keV and temperatures of ~905 °C (No. 3) and ~1750 °C (No. 4). The EDS spectrum of the irradiated surface of both the samples showed a typical pure W spectrum (Table 3). Before the irradiation, a carbon film was on the sample surface in addition to the WC-based carbide layer, but after the irradiation, the presence of carbon was not detected. Most likely, the prolonged exposure to high ion energy and temperature led to the complete carbon sputtering on the tungsten surface.

After testing, the surface of both samples, regardless of the exposure temperature, had a coral-like structure with a lot of helium blisters. Distinctive features of the microstructure of the samples irradiated at different temperatures were that:−the sample retained the grain structure at an irradiation temperature of ~905 °C (Figure 7a) but the grain size increased, and the structure of sample No. 3 itself was very close to the structure of the initial sample after the irradiation at ~1750 °C. We can take the visual uniformity of the grain shade W for some difference.−the grain structure was completely absent on the sample surface at an irradiation temperature of ~1750 °C (Figure 7b). At the same time, unlike samples No. 2 and No. 3, there was no correlation with the orientation of the tungsten grain in this sample, and the structure was characterized by a bumpy structure and vertically directed protrusions over the entire surface.

After irradiation, the tungsten surface with a WC carbide layer had a similar morphology with the initial tungsten irradiated at ~1750°C. A comparative analysis of the results of the initial tungsten and tungsten with a carbide layer showed that the carbide layer based on WC negatively affected the surface properties of tungsten. That is, ~905 ° C was already sufficient for the formation of the observed coral structure with the presence of a carbide layer based on WC. The subsequent increase in the irradiation temperature to ~1750 °C for sample No. 4 led to the predominance of the damaging dose over the grain orientation, and only coral protrusions directed vertically upwards were observed on the surface, which confirmed the low resistance of grains with the orientation W {1 0 3}, {1 0 2}, {4 0 7} and {2 0 3} to some damage during helium plasma exposure. Most likely, we observed a similar evolution of the structure as for sample No. 4 with an increase in exposure temperature or ion energy for the initial sample irradiated at ~1750 °C.

It should be noted that if earlier, after irradiation, the nano-sized holes on the surface of the samples were characterized mainly by spherical shapes on sample No. 1, the appearance of a large number of elongated irregularly shaped pits was observed on sample No. 3 with a carbide layer (WC) (Figure 7a, marked in yellow).

This was caused by the merging of smaller closely spaced pits. That is, we observed a change in the material nanostructure, where the number and concentration of the nanoscale pits increased under the influence of a lower temperature (905 °C) on a sample with a WC carbide layer.

An increase in the irradiation temperature of 1750 °C for sample No. 4 resulted in the appearance of relatively large holes of a regular shape in the form of parallelepipeds (Figure 7b, red arrows). At the same time, the shapes of coral protrusions also changed and acquired a regular shape with clear edges. The presence of spherical meltdowns could be observed at the high points of these protrusions.

This was due to the fact that a small volume of protrusions, taking into account their thermal properties at high temperatures, had a significantly lower thermal conductivity than tungsten [46]. This means that the protrusions heated up more than the main part of the tungsten sample. This resulted in the melting and recrystallization of the protrusions under the influence of temperature and high ion energies, which explains the correct shape of the protrusions with clear edges. Samples No. 2 (initial) and No. 3 (WC) after irradiation were also characterized by this behavior, but the number of such protrusions was very small.

In general, according to preliminary results, the evolution of the damage to the surface structure of the tungsten samples can be described in the following order: No.1 → No.2 →No.3 →No.4, where the main characteristic markers of damage were:−helium blisters appearance;−submicron helium blisters appearance;−increasing the tungsten grain size;−coral surface structure;−increasing the concentration and number of nanoscale pits;−melting and recrystallization of protrusions on the tungsten surface.

### 4.4. Microstructure of the Surface of Tungsten Samples with a Carbide Layer Based on the W_2_C Phase after Irradiation with Helium Plasma

Subsequently, the samples with tungsten carbide based on W_2_C were irradiated with helium plasma at similar parameters. The main difference in the surface microstructure of these samples (Figure 8) was the absence of a coral structure. The nature of the damage in the structure of the samples with a W_2_C-based carbide layer was similar to the structure of the original sample No. 1 irradiated at 905 °C.

It can be seen that the surface of sample No. 5 (Figure 8a) irradiated at ~905 °C and was characterized by a high homogeneity with a uniformly distributed volume content of submicron blisters. The surface itself had a darker shade and relief morphology. The microstructure of sample No. 6 (Figure 8b) irradiated at ~1750 °C was characterized by a more developed morphology with a large number of extended cracks over the entire surface of the sample. It should be noted that after carbidization, the surface was also characterized by a large number of cracks. However, no such cracks were observed in sample No. 5 after the irradiation at a lower temperature. It is possible that the surface layer of the tungsten sample had undergone embrittlement due to the high temperature of the load.

At the same time, at this irradiation temperature, the surface retained a coarse grain structure. However, the identified submicron helium blisters were mostly similar to sample No. 5. At the same time, there was also the presence of a large number of larger polyhedral blisters, which were formed during the connection of smaller submicron blisters. Grains with a darker shade were characterized by the presence of a small number of both large and small helium blisters. However, these grains were characterized by the appearance of holes that were randomly distributed over the surface of the dark grains (Figure 8b, marked in red).

Thus, it can be assumed that the formation of micron and submicron helium blisters on the tungsten surface during the irradiation played an essential role in the formation of the coral structure in the future. The subsequent growth and fusion of these helium blisters could significantly change the shape of the exposed surface and subsurface areas of the tungsten sample.

According to the obtained results, it is clear that changes in the surface structure directly depended on the carbide layer composition. However, in all cases, we observed a complete absence of carbon (Table 3).

Most likely, as it has already been noted, the carbide phases completely sprayed or diffused into the sample depths due to the impact of the high loads. One of the main factors of the tungsten surface resistance to helium exposure is the time required for the decay of the carbide layer. Apparently, with an increase in the irradiation temperature, the W_2_C-based carbide layer decayed at shorter time intervals, which resulted in a more developed damaged surface. However, in order to confirm this statement and subsequently determine the mechanisms of the beneficial effect of this layer on the tungsten surface during the irradiation, it is necessary to conduct experiments with a shorter duration of helium plasma exposure. Moreover, a similar set of statistics is also necessary for the tungsten samples with a carbide layer based on WC, since even here the question of weak surface resistance to exposure to helium plasma in comparison with pure tungsten remains unanswered.

The study of the fine structure sample No. 5 after the irradiation with helium plasma with a W_2_C carbide layer revealed the formation of four layers in depth: a surface layer (≈25 nm); a transitional thin (≈50 nm) layer having a nanocrystalline multilayer structure (Figure 9); an intermediate layer having a submicrocrystalline structure with a crystallite size varying from 150 nm to 250 nm; and the tungsten sample. The formation of helium blisters was detected between the intermediate layer and the tungsten sample. It should be noted that the shape of the blisters found in the work, unlike those known in literary sources, had a more developed morphology and were characterized by a multifaceted shape.

The analysis of the microelectronograms shown in Figure 9 (marked in green) suggested that this foil layer was formed by tungsten crystals. The microelectronogram obtained from the marked region had a quasi-annular structure, which was due to the submicron size of the crystals forming this layer on the one hand, and due to the large size of the selector diaphragm on the other hand. Using the small size of the selector diaphragm, it was determined that the crystallites forming this layer were single crystals. Radial strands on reflexes (indicated by white arrows) indicated the presence of a subgrain structure in a single crystal with a small angle misorientation of the subgrains.

Difficulties arose during the microdiffraction analysis of the nanocrystalline transition layer (≈50 nm). Microelectronograms of this area had a ring structure. In this case, each diffraction ring can be represented by the superposition of several diffraction rings with close interplane distances. The latter may mean the presence of a certain number of phases in the surface layer at the same time, or the presence of one phase having a variable crystal lattice parameter, which allows one to assume the presence of another phase in this layer besides tungsten.

To solve the problem with the ambiguity in determining the phase composition, electronograms were obtained from nanosized sections of the foil from the surface layer, indicated in Figure 9 by numbers. The analysis of electronograms revealed the reflexes of tungsten and tungsten carbide with a W_2_C composition. Thus, the surface layer had a multiphase nanocrystalline structure.

## 5. Conclusions

To summarize, based on the analysis of the results presented in this paper, a number of conclusions can be drawn about the influence of the carbide layer on plasma interaction processes with tungsten.

Experiments were carried out on the effect of helium plasma on the surface of tungsten and tungsten with a carbide layer at temperatures of ~905 °C and ~1750 °C. The concentration, electron temperature, ion flux and fluence of helium ions under the tungsten irradiation were estimated. The ion fluence under the irradiation was 7.48·10^25^ ÷ 8.26·10^25^ ion/m^2^, which is in line with the expected parameters in ITER.

The tungsten surface irradiation with the ion energy of 2 keV resulted in strong damage to the tungsten surface and sputtering. The similar nature of the surface morphology due to the irradiation with pure He+ ions at higher temperatures was expected and it was consistent with other studies.

The presence of a carbide layer during the interaction of helium plasma with tungsten led to a relatively rapid sputtering of carbon. In accordance with the results of studies on the fine structure of the surface layer of tungsten, the diffusion of carbon into the depth of the sample was found.

The presence of a carbide layer in the interaction of helium plasma with tungsten resulted in carbon sputtering as well as its part diffusion into the sample depth in accordance with the results of studying the fine structure of the surface layer.

According to the results obtained, the WC-based tungsten carbide layer had a weaker resistance to helium plasma damage than the initial tungsten and tungsten with a W2C carbide layer. An increase in the temperature of the plasma exposure resulted in more severe damage to the surface and the formation of melted and recrystallized protrusions.

According to the obtained results, the W2C-based tungsten carbide layer had a higher resistance to the helium plasma exposure, which persisted even with an increase in the exposure temperature. However, high temperature exposure at 1750 °C resulted in the surface layer embrittlement.

A microstructural analysis revealed that the carbide layers negatively affected the surface properties of tungsten, accelerating the formation of a coral structure at low irradiation temperatures and leading to significant changes in the tungsten surface morphology.

Further research will focus on the effect of carbide layers on the tungsten surface as a potential diffusion barrier, which plays an important role in the capture, retention and diffusion of helium and hydrogen ions into tungsten. In situ tungsten thermodessorption (TDS) experiments will be carried out to obtain new data on the influence of carbide layers on plasma–surface interaction.

## 6. Patents

Patent for the invention “Method for Producing Tungsten Carbides in Plasma-Beam Discharge”, Authors: Skakov M.K., Sokolov I.A., Batyrbekov E.G., Tulenbergenov T.R., Miniyazov A.Zh. No. 34269 dated 04/09/2020 RSE “National Institute of Intellectual Property” of the Ministry of Justice of the Republic of Kazakhstan, 2020.

## Figures and Tables

**Figure 1 materials-15-07821-f001:**
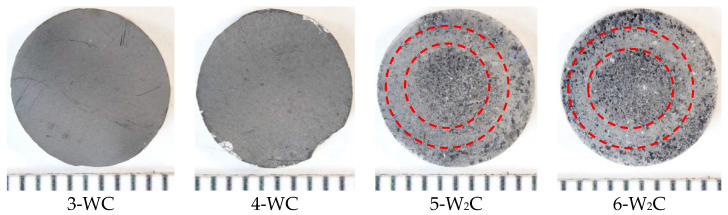
The appearance of tungsten samples after carbidization experiments.

**Figure 2 materials-15-07821-f002:**
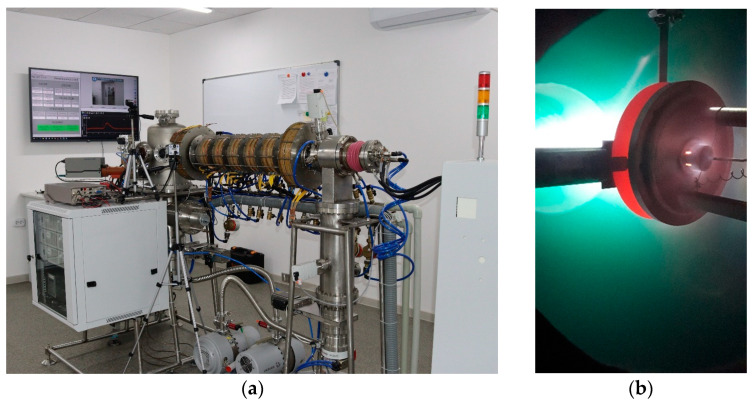
Plasma beam installation (**a**) and the back side of the sample (**b**) in the chamber.

**Figure 3 materials-15-07821-f003:**
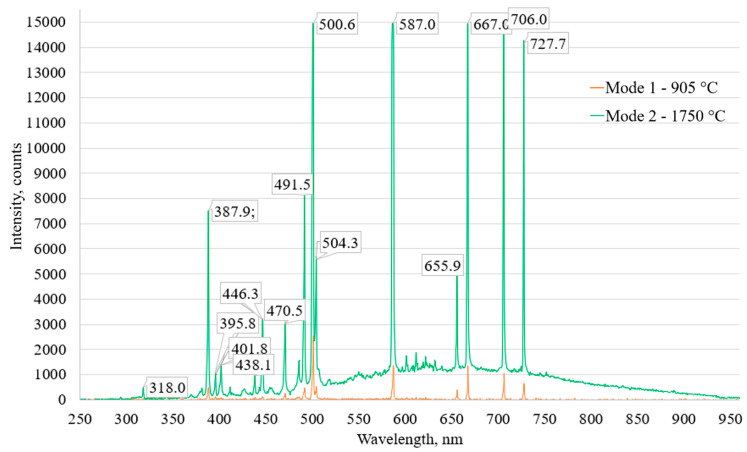
Graph of optical spectra under two temperature modes (905 °C and 1750 °C) of tungsten samples irradiation with helium plasma.

**Figure 4 materials-15-07821-f004:**
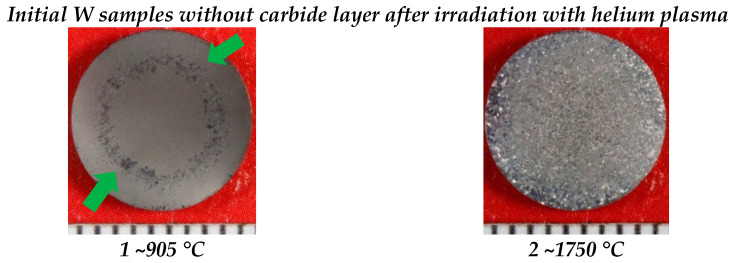
Surface of the initial and with carbide-layer tungsten samples after irradiation with helium plasma at temperatures of ~905 °C and ~1750 °C.

**Figure 5 materials-15-07821-f005:**
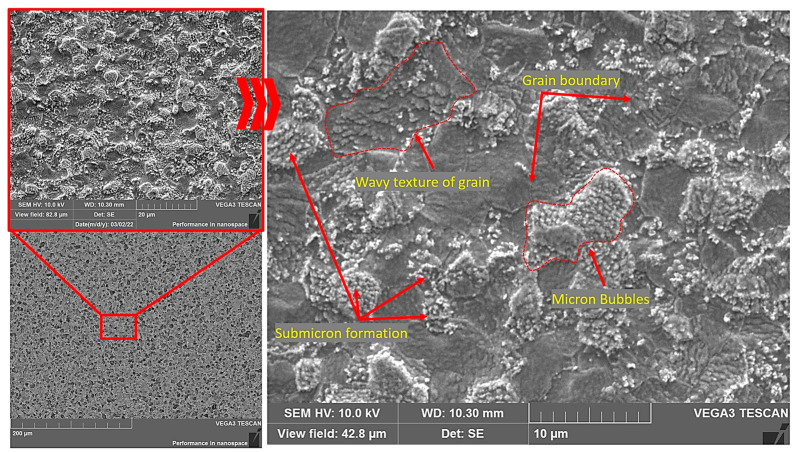
Microstructure of sample No. 1 surface after irradiation with helium plasma at ~905 °C.

**Figure 6 materials-15-07821-f006:**
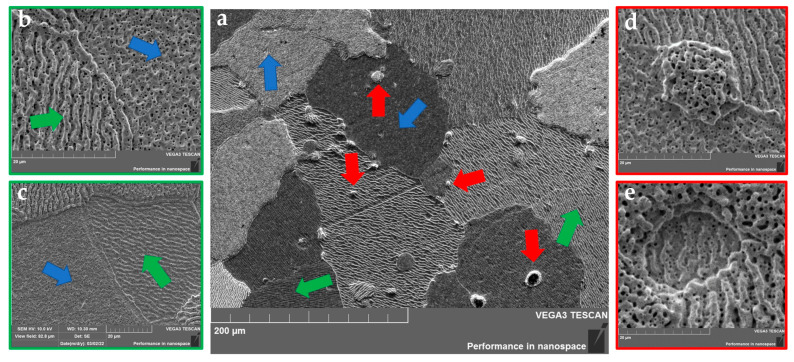
Microstructure of sample No. 2 surface after irradiation with helium plasma at ~1750 °C (red arrows indicate formed helium blisters, blue arrows indicate grains with extended oriented protrusions, green arrows indicate grains with a bumpy structure). (**a**)—coral structure; (**b**,**c**)—changes in morphology in grains; (**d**,**e**)—helium blisters.

**Figure 7 materials-15-07821-f007:**
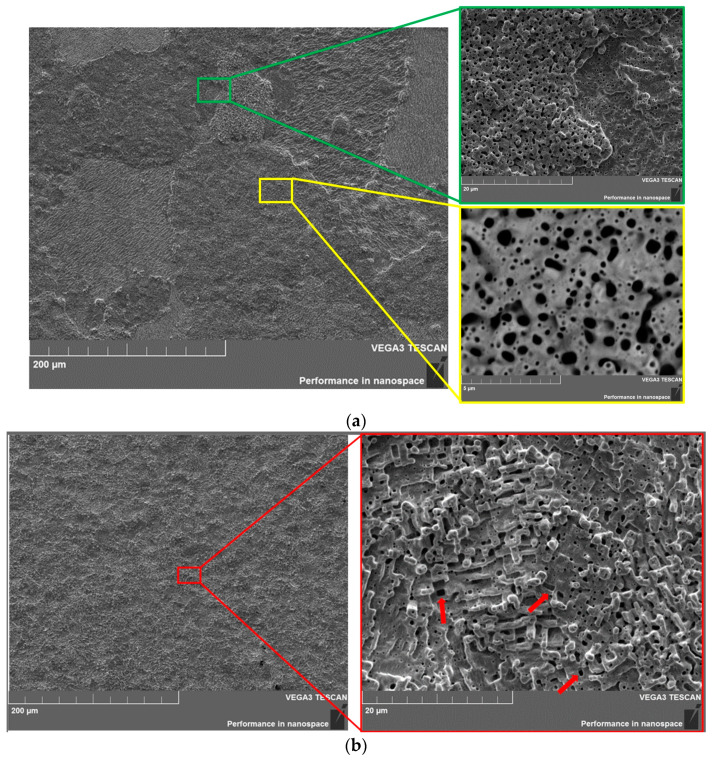
Microstructure of the surface of tungsten samples with a carbide layer based on the WC phase after irradiation with helium plasma at different temperatures (red arrows indicate large pores formed during irradiation). (**a**) Sample No. 3 **~905 °C**, (**b**) sample No. 4 **~1750 °C**.

**Figure 8 materials-15-07821-f008:**
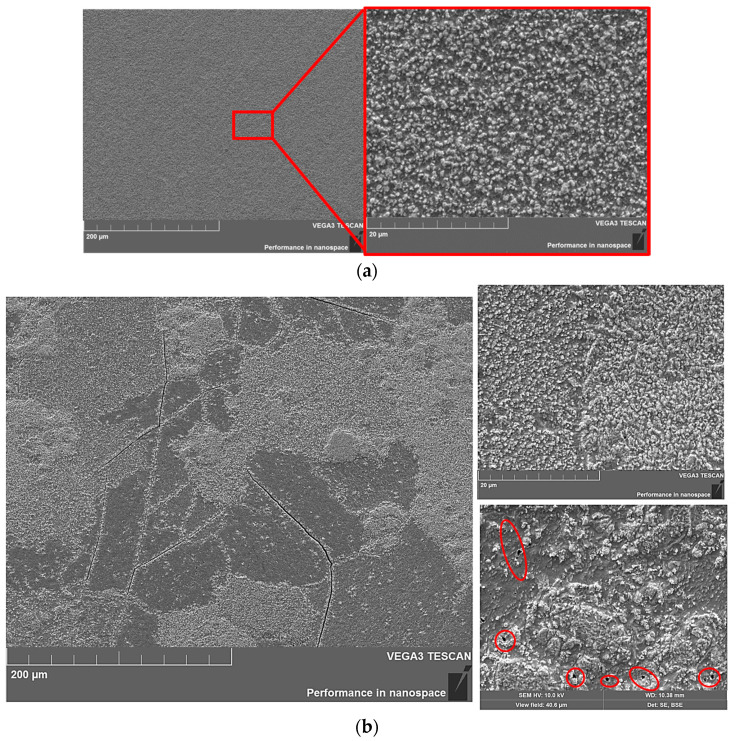
Microstructure of the surface of tungsten samples with a W_2_C carbide layer after irradiation with helium plasma at different temperatures (the locations of pores is marked in red). (**a**) Sample No. 5 **~905 °C**, (**b**) sample No. 6 **~1750 °C**.

**Figure 9 materials-15-07821-f009:**
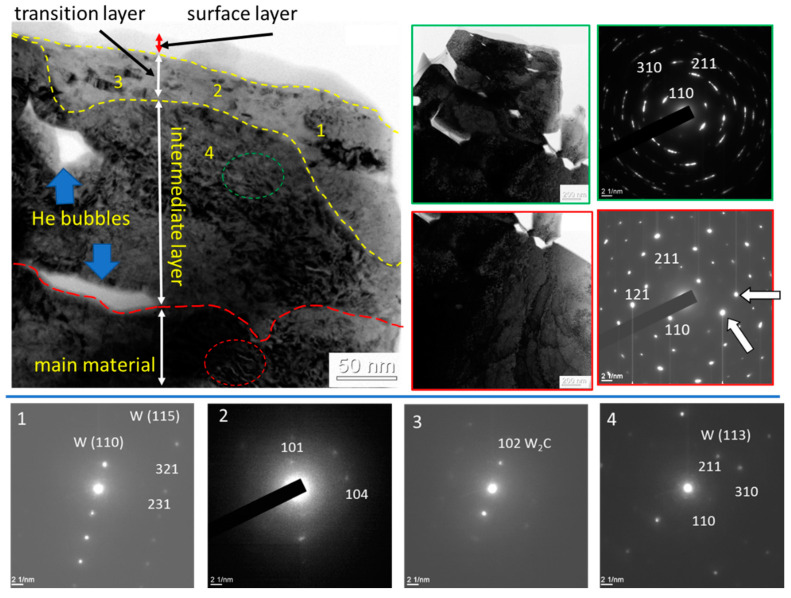
An electron microscopic image of the nanostructured surface layer of sample No. 5; a is a light field; 1–4 areas marked with red and green are microelectronograms obtained from these areas with numbers 1–4. Area No.1-W; No.2-W + W_2_C; No.3-W + W_2_C; No.4-W; green boundary, reflex [110]W (bcc lattice, a = 0.3158 nm); red boundary, reflex [211]W (plane of 113 type).

**Table 1 materials-15-07821-t001:** Parameters of experiments on the effect of helium plasma on tungsten.

Sample No	Surface Condition of Tungsten	Electron Beam Power, W	Ionic Current on the Sample, mA	Front Side Temperature, °C
**1 (W)**	Without carbide layer (initial)	470	78	905 ± 10
**2 (W)**	Without carbide layer (initial)	1635.1	100	1750 ± 10
**3 (WC)**	With WC carbide layer	335.4	27	905 ± 10
**4 (WC)**	With WC carbide layer	1050	66	1750 ± 10
**5 (W_2_C)**	With W_2_C carbide layer	322.5	28	905 ± 10
**6 (W_2_C)**	With W_2_C carbide layer	1093.4	79	1750 ± 10

**Table 2 materials-15-07821-t002:** Parameters of helium plasma depending on irradiation modes.

Sample	Surface Temperature, °C	Ion Concentration, m^−3^	Electronic Temperature, eV	Ion Flux, Ion/m^2^s	Ion Fluence, m^−2^
**1 (W),** **3 (WC),** **5 (W_2_C)**	905	2.78·10^18^	7.39	7.645·10^21^	8.26·10^25^
**2 (W),** **4 (WC),** **6 (W_2_C)**	1750	2.52·10^18^	7.69	1.1·10^21^	7.48·10^25^

**Table 3 materials-15-07821-t003:** Results of EDS analysis of the surface of tungsten samples.

Sample	C	O	W	Total
Samples before irradiation, mass. %				
1 (W)	-	-	100.00	100.00
2 (W)	-	-	100.00	100.00
3 (WC)	98.43	0.35	1.21	100.00
4 (WC)	97.18	1.07	1.76	100.00
5 (W_2_C)	30.54	1.49	67.97	100.00
6 (W_2_C)	31.31	1.36	67.33	100.00
Samples after irradiation with helium plasma, mass. %				
1 (W)	-	6.75	93.25	100.00
2 (W)	-	5.60	94.40	100.00
3 (WC)	-	7.60	92.40	100.00
4 (WC)	-	-	100.00	100.00
5 (W_2_C)	-	8.59	91.41	100.00
6 (W_2_C)	-	-	100.00	100.00

## Data Availability

The data used to support the findings of this study are available from the corresponding author upon request.

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
