# Peer review of "Influence of the Carbidized Tungsten Surface on the Processes of Interaction with Helium Plasma"

_materials, 2022, doi:10.3390/ma15217821_

Round 1

Reviewer 1 Report

The work present the results on the changes on morphology and structure of W carbidized samples exposed to helium irradiation under ITER loads conditions.

The paper bring some moderate results and can be published  afetr major revision.

See attached file

Author Response

Thank you very much for your comprehensive comments.

Please find in attached file our answer and corrects.

Reviewer 2 Report

The paper shows a relevant approach to applications in nuclear fusion and thus can be accepted as is

Author Response

Thank you very much for your kind support.

English of the paper was improved.

Reviewer 3 Report

this work focused on the interaction between helium plasma and tungsten based divertor materials under the simulated condition to that of nulcear reactors, which are of extreme importance to the future possible application of nuclear reactor, and provide useful and reliable data and documents. I strongly recommend it for publication in the present form. 

Author Response

Thank you very much for your kind support.

Reviewer 4 Report

please see the attached pdf

Author Response

Thank you very much for your support.

Please find in attached file our answer and corrects.

Round 2

Reviewer 1 Report

The Authors completed the review  as I asked . The manuscript can be published